# The Research of Crystalline Morphology and Breakdown Characteristics of Polymer/Micro-Nano-Composites

**DOI:** 10.3390/ma13061432

**Published:** 2020-03-21

**Authors:** Yujia Cheng, Guang Yu, Xiaohong Zhang, Boyang Yu

**Affiliations:** 1Mechanical and Electrical Engineering Institute, University of Electronic Science and Technology of China, Zhongshan Institute, Zhongshan 528400, China; chengyujia1068@163.com; 2Key Laboratory of Engineering Dielectrics and Its Application, Ministry of Education, Harbin University of Science and Technology, Harbin 150080, China; x_hzhang2002@hrbust.edu.cn; 3Mechanical and Electrical Engineering Institute, Zhongshan Polytechnic, Zhongshan 528400, China; 0201965001@zspt.cn

**Keywords:** micro-nano-composites, interfacial structure, crystalline morphology, breakdown characteristics

## Abstract

In this article, low-density polyethylene (LDPE) was used as a matrix polymer, the Micro-ZnO and Nano-ZnO particles were used as the inorganic filler. With the melt blending method, the Nano-ZnO/LDPE(Nano-ZnO particles doping into LDPE), Micro-ZnO/LDPE(Micro-ZnO particles doping into LDPE) and Micro-Nano-ZnO/LDPE (Nano-ZnO and Micro-ZnO particles doping into LDPE in the same time) composites were prepared. Then, the inorganic filler and the composites were dealt with structural characterizations and analysis by Fourier transform infrared (FTIR), Polarization microscope (PLM), and Differential scanning calorimeter (DSC). Besides, these samples were dealt with (alternating current) AC breakdown performance test. The micro-experimental results showed that the Micro-ZnO and Nano-ZnO particles doping reduced the crystal size and increased the crystallization rate. With the change of cell structure, the crystallinity of composites increased. The crystallinity order of different samples was as follows: LDPE < Micro-ZnO/LDPE < Nano-ZnO/LDPE < Micro-Nano-ZnO/LDPE. From the breakdown of the experimental result, with the same mass fraction of the different inorganic doping of particles, the breakdown strength of these composites was different. The Nano-ZnO particle doping could improve the breakdown strength of composites effectively. Among them, the breakdown strength of Nano-ZnO/LDPE and Micro-Nano-ZnO/LDPE were 11% higher and 1.3% lower than that of pure LDPE, respectively. Meanwhile, the breakdown strength of Micro-composite was the lowest but its Weibull shape coefficient was the highest. Therefore, the Micro-ZnO doping was helpful for the Nano-ZnO dispersing in the matrix, which produced the Micro-Nano-synergy effects better.

## 1. Introduction

With economic growth, the power supplies were far from enough to meet the growing demand for economic and social development. Therefore, the greater demands for transmission capacity and safety of the power grid had been put forward [1,2]. Due to the conveying distance limit of the high-voltage transmission system, the cross-regional power exchange and new energy access were difficult to achieve. Besides, some problems became more serious, such as the lack of transmission corridors and the excess of short-circuit current. (Ultra High Voltage) UHV transmission possessed many advantages such as low-loss, long-distance, and high-capacity, which could offset the flaw of current high voltage transmission. Therefore, it was the inevitable development trend for the power grid [3,4]. With the improvement of transmission voltage degree, the higher electrical insulation performance request to insulating material of transmission line and power equipment were proposed. The polymer materials were widely used in the electrical insulation field for their fine electrical insulation and processing performance. Meanwhile, the heat resistance and insulation characteristic could not be ignored in high insulation level power equipment. Domestic and foreign scholars had done lots of researches on it. The main research was based on doping certain additives into polyolefin material. Besides, the different preparation and characterization methods were used, from which the high dielectric properties insulation material was developed [5]. The results showed that the microparticles doping into polymer could improve the thermal conductivity of composites but degrade its electrical insulation performance [6]. The appearance of Nano-composite made up for the shortage of conventional polymer material. The nanoparticles possessed a large specific surface area and surface energy, which would form the interaction zone with polymer matrix, that was, the interface microdomain. The presence of interface microdomain could limit the movements of polymer molecule chains around nanoparticles, which improved the mechanical properties of composites [7,8,9]. Besides, the potential barrier of interface microdomain was high, in which lots of deep level traps existed. These deep traps could capture the charges; then the charge transfer was limited. Therefore, the thermionic was hard to be excited, which improved the insulation properties of composites. Research showed that the moderate nanoparticle doping onto the polymer could improve the space charge distribution and breakdown field strength [10,11,12,13,14,15]. Besides, the partial discharge reduced, and the resisting electrical tree was improved. On the other hand, the high surface energy of nanoparticles would cause the agglomeration easily [16,17,18,19]. Therefore, the Micro-Nano-composite combined the merits of microparticles and nanoparticles doping, respectively, and this composite had become a new direction in polymer composite.

In this article, the inorganic microparticles and nanoparticles were doped into the polymer matrix simultaneously. The Micro-Nano-composite was prepared by the melting blending preparation process. The inorganic ZnO particles were a kind of wide bandgap compound semiconductor material; hence, these nonlinear characteristic inorganic semiconductor material being added into the polymer material could homogenize the electric field distribution and improve the dielectric strength of the insulation structure. Therefore, the Micro-ZnO/LDPE and nano-ZnO/LDPE composites were chosen for the study object in this article. Firstly, the crystalline structure of the composites was tested by FTIR, PLM, and DSC, from which the effect of microparticles and nanoparticles doping to the crystalline morphology of polymer composites could be explored. Then, the electric field strength resistance of different samples could be tested by power frequency AC breakdown. Finally, the effects of inorganic particle doping on the structure and insulation characteristic of polymer composites were studied, from which the mechanism of Micro-Nano-synergy effects on the dielectric properties of composites could be explored further.

## 2. Experimental Method

### 2.1. Main Raw Materials and Instrument

Experimental materials: Low density polyethylene(LDPE), of which met index was 1.5 g/10 min; Micro-ZnO particles and Nano-ZnO particles, of which size were 40 nm and 1 μm, respectively; *γ*-Methacyloxypropyl Trimethoxylsilane (MATS) Silane coupling agent KH-570; Antioxidant 1010; The 99% pure P-xylene; 5% concentration Potassium permanganate and Concentrated sulfuric acid.

The information of the device’s names, models, and manufacturers were shown in Table 1.

### 2.2. Preparation of Composites

In this article, the ZnO particles were dealt with surface modification by the silane coupling agent. According to the hydrolysis reaction, the silane coupling agent produced the reaction silanol. After the dehydration condensation reaction, it formed the oligomer structure and reacted with -OH in the surface of ZnO particles. Therefore, the stronger bonding was formed between molecules. The organic and inorganic group existed in the coupling agent simultaneously, the nanoparticle surface could be covered by coupling agent with a physical and chemical reaction. The purpose of inorganic particles modification was realized, and the ZnO particles after surface modification could be combined with polymer matrix better [20].

The ZnO particle modification should follow these steps: Firstly, the ZnO powder should be placed in the oven to dry for 24 h. A certain amount of ZnO power was added into the mixture of absolute and water. After the ultrasonic vibration for 1 h, the mixture was poured into the three-necked flask. Secondly, the three-necked flask was placed in the 85 °C thermostat water bath and dealt with power stir for 2 h. After the uniformity stir, the silane coupling agent was added into the mixture by dropwise. Then the mixture was dealt with many filtration and washing by sand core funnel and filter flask. After drying and grinding, the ZnO particles after surface modification were obtained.

In this article, the (LDPE) was used as a matrix polymer. The size of Micro-ZnO and Nano-ZnO particles was 1 μm and 40 nm, respectively. The filler content of three composites were shown in Table 2. The Micro-ZnO/LDPE, Nano-ZnO/LDPE, and Micro-Nano-ZnO/LDPE composites with different filler content were prepared by melting a blend with LDPE in a torque rheometer. The mixing temperature was set to 150 °C, and the screw speed was 40 r/min. After the process of mixing, all these samples were rolled into sheets by plate vulcanizing press machine. The molding temperature was set to 150 °C and the pressure was 10 Mpa. Then, these samples were cooled under the pressure of 15 MP. Finally, the test sample preparation was completed [21].

### 2.3. Structure Characterization and Performance Testing of Composite

In the FTIR experiment, the KBr was used as background, and the samples were scanned in the wavenumber region of 4000–500cm^−1^.

The experimental samples were observed by PLM. Firstly, these samples were placed into a certain concentration of potassium permanganate and concentrated sulfuric acid. After a period of time for corrosion, these samples were taken out and placed in an ultrasonic machine for surface cleaning. Then, all these samples were placed on the glass slide, from which the samples’ crystalline morphology could be observed.

The experimental samples were tested by DSC. Under the protection of drying nitrogen at a flow rate of 150 mL/min, the temperature of DSC rose uniformly in 10 °C/min until 150 °C, then dropped to room temperature at equal rate. The enthalpy change in the cooling process could be recorded, and the crystallization temperature T_c_ and the width of exothermic crystallization peaks ΔT_c_ could be calculated. After that, the test temperature of DSC rose uniformly to 150 °C again. The enthalpy change in the heating process was recorded, from which the melting enthalpy ΔH_m_ and melting temperature T_m_ were calculated.

The power frequency AC system was used for the breakdown of the property test. The thickness of the samples was 100 μm, and the breakdown field strength was calculated by Equation (1).
(1)E=U/d

From Equation (1), U was the voltage, and *d* was the sample thickness of breakdown points. The breakdown data were analyzed by MINITAB, from which the Weibull distribution breakdown field strength E and shape parameter β of different samples could be calculated. β reflected the dispersion of breakdown data, and *E* reflected the degree of breakdown field strength.

In these experiments, the size of Micro-ZnO and Nano-ZnO particles were 40 nm and 1 μm, respectively. The mass fraction of inorganic ZnO particles in each composite was 1%, 3%, and 5%, respectively. According to the analysis of the breakdown property test, the Weibull distribution relationship diagram between breakdown field strength E and shape parameter β of Micro-ZnO/LDPE, Nano-ZnO/LDPE, and Micro-nano-ZnO/LDPE composites could be made. After that, the breakdown property of different ZnO/LDPE composites could be explored.

## 3. Results and Analysis

### 3.1. FTIR Characterization of Composites

The FTIR patterns of ZnO particles before and after surface modification by the silane coupling agent is shown in Figure 1.

From Figure 1, the hydroxyl (-OH) number of ZnO particles after surface modification decreased evidently, which illustrated the silane coupling agent had reacted with the hydroxyl on the ZnO surface. The silane molecules could be grafted on the surface of the ZnO particles, and the surface coating was formed. At this time, the lipophilic group existed on the surface of ZnO particles. Then the nanoparticles would be combined with the polymer matrix better. According to the experimental result of chemical titration, the hydroxyl number in ZnO particles surface before and after surface modification were 15 and 5 per square nanometer. The characterization result was consistent with chemical titration calculations.

The infrared spectrum of LDPE, Nano-ZnO/LDPE, Micro-ZnO/LDPE, and Micro-Nano-ZnO/LDPE were shown in Figure 2.

From Figure 2, there were three strong absorption peaks at the wavelength of 2850 cm^−1^, 1450 cm^−1^, and 723 cm^−1^ in FTIR patterns of all tested samples. They were C-H bond stretching vibrational absorption peak, C-H bond in-plane bending vibrational absorption peak, and C-H bond out-plane bending vibrational absorption peak, respectively. This illustrated that the ZnO particles doping onto composites did not change the position of characteristic peaks in the matrix LDPE. In other words, the ZnO did not change the molecular chain structure of LDPE. In the wavenumber close to 3500 cm^−1^, the -OH vibrational peak of Micro-ZnO and Nano-ZnO particles almost disappeared. This was because the Micro-ZnO and Nano-ZnO particles were coated by large molecules of LDPE and could not be detected during the preparation of Micro-ZnO/LDPE and Nano-ZnO/LDPE composites. Besides, on the surface of ZnO particles, -OH combine with -H to form the water molecules and evaporate from materials during the preparation of composites. At last, the new absorption peaks existed at the wavelength of 1140 cm^−1^ and 1200 cm^−1^ in FTIR patterns of composites. They were the methylene symmetry and asymmetry low-intensity bending vibrational absorption peak, and the C-H bond bending vibrational absorption peak which were introduced by the ZnO filler [22].

### 3.2. DSC Characterization of Composites

In order to further explore the effect of microparticles and nanoparticles doping to LDPE crystallization, the different samples were dealt with DSC characterization, from which the heating process and cooling process curve could be obtained. They were shown in Figure 3. After the calculation, the melting peak and crystallinity of different samples were shown in Table 3. Because the melting points of inorganic ZnO-particles were higher than which of pure LDPE, the pure LDPE was used as the matrix. When the ZnO/LDPE composites were dealt with the DSC test, the thermomechanical behavior of LDPE was similar to which of composites.

According to the DSC test, the effect of different mass fraction of Micro-ZnO and Nano-ZnO doping on the crystalline morphology and melting enthalpy of LDPE could be explored. According to the DSC heating process curve of pure LDPE and composites, the melting enthalpy ΔH_m_ and melting peak temperature T_m_ of different samples could be obtained. Finally, according to Equation (2), the materials crystallinity X_c_ could be calculated by melting enthalpy ΔH_m_ and the mass fraction of inorganic particles ω [23,24]. In this experiment, the temperature error was ± 0.1 °C.
(2)$$X_{c}=\frac{\Delta H_{m}}{\left(1-\omega \right)H_{0}}\times 100\%$$

In Equation (2), ΔH_m_ is the melting enthalpy (J/g). H_0_ is the melting enthalpy under holocrystalline, which was 293.6 J/g for LDPE. *ω* is the ZnO mass fraction in the composite. T_mc_ is the initial melting temperature.

From the experiment result of Figure 3 and Table 3, the melting peak temperature T_m_ of pure LDPE was lower than which of both Nano-ZnO/LDPE and Micro-ZnO/LDPE composites. It illustrated that the samples would crystallize at higher temperatures with the Micro-ZnO or Nano-ZnO particles doping. The width of exothermic crystallization peak ΔT_c_ of Micro-ZnO and Nano-ZnO composites was shorter than which of pure LDPE. Besides, the melting peak temperature T_m_ of Micro-Nano-ZnO/LDPE was higher than which of pure LDPE. The Micro-Nano-ZnO particles doping played the role of the nucleating agent, which advanced the ordered arrangement of the LDPE molecular chain. Therefore, the crystallinity of all ZnO/LDPE composites was higher than which of pure LDPE. At the same time, there was a single exothermic peak in the DSC curve of all samples, which illustrated that the previous crystalline changes of LDPE did not change after the Nano-ZnO and Micro-ZnO doping. The ZnO was a kind of inorganic materials, which provided good heat resistance. Besides, the inorganic particles provided excellent thermal conductivity, from which the heat would be transferred quickly. Therefore, the melting temperature of the composites were higher [25,26]. From the experimental results, the crystallinity of composite N3M2 was the highest. This was because the macromolecule chains of pure LDPE existed around the ZnO particles. When the number of ZnO particles increased, the proportion of crystallization per volume in the polymer was higher. Under the same mass fraction, the number of inner particle order of different samples was as follows: N5 > N3M2 > M5. However, the Nano-ZnO particles agglomeration in the matrix resin weakened the effect of heterogeneous nucleation, so the crystallinity order of different samples was as follows: N3M2>N5>M5. In conclusion, the Micro-ZnO and Nano-ZnO particles doping could improve the crystallinity and crystallization rate of LDPE. The polymer crystalline structure was changed, and the macroscopic polymer was affected.

### 3.3. PLM Characterization of Composites

To further characterize the effect of Micro-ZnO and Nano-ZnO particles doping on the crystallization of LDPE, the PLM patterns of LDPE and different ZnO/LDPE composites were shown in Figure 4.

From Figure 4, the crystal size of pure LDPE was bigger. Between the grains, the non-crystalline regions were larger, and the arrangement of the grains was loose. After the ZnO particles doping, the grain size of the composites decreased and the arrangement of the grains was regular and close. Therefore, the ZnO particles doping played the role of heterogeneous nucleation agents. When the same mass fraction of ZnO particles were all 5%, the crystalline morphology of M5, N5, and N3M2 composites was different. The crystal size order of different samples’ interior was as follows: N5 < N3M2 < M5.

According to the analysis of PLM patterns, the surface activity of inorganic Nano-ZnO was high, and the interaction of Nano-ZnO with polythene molecular chains was strong. Therefore, the Nano-ZnO particle doping was equal to the heterogeneous nucleation agent, which affected the crystallization behavior of composites. In semi-crystalline polymer-based composites, the inorganic filler is the heterogeneous nucleation point of the polymer matrix crystallization. The crystal growth of polymer molecule on the surface of filler would present two typical interfacial crystalline morphologies, which were transcrystalline and hybridization shish-kebab [27,28]. Based on the crystallization kinetics theory, the structure model of LDPE, Nano-ZnO/LDPE, Micro-ZnO/LDPE, and Micro-Nano/LDPE composites are shown in Figure 5.

LDPE is the semi-crystalline polymer; therefore, the crystalline region and non-crystalline region went together in this sample. They were homogeneous nucleation crystalline region and amorphous region, of which concrete structure are shown in Figure 5a. The PLM observation showed tight bonding between the nanoparticles and the polymer matrix. In general, the normal semi-crystaline polymer doping existed in non-crystalline regions preferentially. For the Nano-ZnO, it possessed a large specific surface area and had strong adsorption. So, the polymer chains could be adsorbed to the surface of nanoparticles, which acted as the physical crosslinking points. At the same time, according to the surface modification of nanoparticles by the coupling agent, there was a lipophilic group on the surface of the ZnO particles. The Nano-ZnO particles could be adhered with the polyethylene matrix tightly by covalent bond; therefore, the limited polymer chains would arrange around the nanoparticles in the way of radiating, which was similar to the model in Figure 5b. For the Micro-ZnO, only a common electrostatic force existed between the ZnO-particles and the polymer matrix, so the interfacial adhesion was relatively weak. Therefore, the microparticles were introduced by a steric effect mainly, from which the homogeneous crystal cell would develop around the polymer molecular chain. The structural model was shown in Figure 5c. In preparation for the Micro-Nano-composites, the Nano-composite with the content of 3% ZnO was used as the matrix. Then the Micro-ZnO particles were added into the matrix until its content reached 2%. The inorganic phase of Nano-ZnO composite dispersed uniformly. Besides, the polymer chains bonding around Nano-ZnO particles were affected by the steric hindrances of Micro-ZnO particles. Therefore, the ordering degree of segments arrangements would decrease, and the thickness of the bonding layer was thinned. The crystalline morphology of Micro-Nano-composite was the mixed mode of molecular chains parallel arrangement and radial arrangement, which was shown in Figure 5d.

Combining the experimental results of PLM with DSC, the nanoparticles and microparticles doping played the role of nucleating agent, which promoted the ordered arrangement of LDPE molecule chains in a different way. The macromolecule chains would develop around the inorganic ZnO particles.

### 3.4. Breakdown Test of Composites

The Weibull distribution curve of breakdown field strength in LDPE, different Micro-ZnO/LDPE, and Nano-ZnO/LDPE composites were shown in Figure 6.

From Figure 6, the breakdown field strength of Nano-ZnO/LDPE composites with different mass fraction was different. Among them, the breakdown field strength of Nano-ZnO/LDPE with a 3% mass fraction was the highest and 11% higher than which of pure LDPE. While the mass fraction of Nano-ZnO was 1% and 5%, the breakdown field strength was relatively low. Combined with the result of crystalline morphology and crystallinity parameters, the trends of breakdown field strength had good consistency with the crystallization scale. According to the free volume breakdown theory, the free volume showed the spatial scale condition of electron accumulating kinetic energy under an electric field. When the free volume increased, the electrons average free path was longer, the chances of ionization collision were higher, and the breakdown field strength was lower [29,30]. When the mass fraction of Nano-ZnO was 1%, the heterogeneous nucleation effect of Nano-ZnO was not obvious. The crystal grain size of composites existed a certain dispersion, and the scale of the non-crystalline region was larger. When the mass fraction of Nano-ZnO was 5%, owing to the agglomeration of the nanoparticles, the crystallization scale was non-uniform, and the free volume was larger. Therefore, the breakdown field strength of Nano-ZnO composites with 1% and 5% mass fraction were higher than which of pure LDPE, but lower than which of Nano-ZnO composites with a 3% mass fraction. From the crystallization properties of Nano-ZnO/LDPE with 3% mass fraction, the crystallization scale was the smallest. Besides, the grains arrayed densely, and the scale of the non-crystalline region was relatively small. Therefore, the breakdown field strength of Nano-ZnO composites with 3% was the highest.

In addition, from Figure 6, the breakdown field strength of Micro-composites with different mass fraction was different. The breakdown field strength of all micro-composites was lower than which of pure LDPE. As the mass fraction of microparticles increased, the breakdown field strength of the composite decreased gradually. From the experimental results, when the mass fraction of Micro-ZnO particles was 1%, 3%, and 5%, the breakdown field strength of the ZnO/LDPE composites was 15%, 19.5%, and 22.4%, lower than that of pure LDPE. This was because the microparticles were different from nanoparticles, which did not possess the special effects, such as small size effect and large specific surface area. Besides, there was a weak intermolecular force between microparticles and polymer matrix. It was equivalent to add the impurities into LDPE, from which the segment structure of polymer macromolecule chains was blocked. These flaws were fatal in the process of the electric field explosion. In conclusion, the breakdown field strength of Micro-ZnO/LDPE composites decreased.

The Micro-Nano-ZnO/LDPE composites with a 5% mass fraction were used for breakdown tests. In the test samples, the Micro-ZnO and Nano-ZnO particles were added with different proportions, which was denoted by N1M4, N2M3and N3M2. The breakdown test result was shown in Figure 7.

From Figure 7, the breakdown field strength of all the Micro-Nano-ZnO/LDPE was lower than which of pure LDPE. As the mass fraction of Nano-ZnO particles increased, the breakdown field strength of Micro-Nano-/LDPE increased gradually. On the contrary, when the mass fraction of Micro-ZnO particles increased, the breakdown field strength of Micro-Nano-ZnO/LDPE would decrease. The breakdown field strength of N1M4, N2M3, and N3M2 were 15.7%, 3.8%, and 1.3% lower than which of pure LDPE, respectively. This was because the deep traps were introduced by nanoparticle doping [31,32], and the close interfacial structure was formed by the covalent bond of nanoparticles with the polyethylene matrix. The non-crystalline region free volume decreased, and the conductive path was complicated, which would prolong the sample service life. Besides, the interfacial binding strength between microparticles and polyethylene matrix was weaker. When the applied electric increased, the microparticles were as the impurity and introduced the traps, which weakened the effect of nanoparticles. Therefore, the breakdown field strength of Micro-Nano-ZnO/LDPE decreased. On the other hand, the microparticles possessed superior mechanical properties and thermal conductivity, which would decrease the heat accumulation in Micro-Nano-ZnO/LDPE. As the number of thermionic decreased, the probability of thermionic destroying the polymer matrix would reduce. In summary, when the mass fraction of nanoparticles and microparticles was proper, the breakdown field strength of Micro-Nano-ZnO/LDPE was relatively high.

## 4. Conclusions

In this article, the LDPE was used as the matrix, the Micro-ZnO and Nano-ZnO particles were used as the inorganic filler. According to the melt blending, the Micro-ZnO/LDPE, Nano-ZnO/LDPE, and Micro-Nano-ZnO/LDPE were prepared. According to the FTIR, PLM, and DSC test, the microstructure and crystalline morphology of composites were characterized. Besides, the composites were dealt with a breakdown test. The conclusions were as follows:(1)From the test result of FTIR, the chemical reaction happened between the organic nanoparticles and the coupling agent. Besides, the peaks value of composites in FTIR was lower than which of pure LDPE. From the test result of PLM, the inorganic ZnO-particles doping played the role of a nucleating agent, which would reduce the composite crystal size. The number of nucleation and the interfacial structure of the non-crystalline region increased obviously. At the same time, the grain arrangement was close inside the media.(2)From the test result of DSC, the ZnO particles doping improved the crystallinity and crystallization rate of pure LDPE. Among them, the crystallinity of Micro-Nano-ZnO/LDPE was the highest, and nano-ZnO/LDPE took the second. On the other hand, the melting temperature and crystallization rate of Micro-ZnO/LDPE was the highest. Based on the analysis of crystallization kinetics theory and combining the experiment result of PLM and DSC, the crystalline morphology model of Micro-Nano-ZnO/LDPE was built, and the mechanisms of Micro-ZnO and Nano-ZnO particles doping to the crystallization properties of the polymer were explored.(3)From the breakdown test result of Micro-ZnO and Nano-ZnO composites, the breakdown field strength of Nano-ZnO/LDPE was the highest and 11% higher than which of pure LDPE. The breakdown field strength of Micro-ZnO/LDPE was the lowest and 15% lower than which of pure LDPE. The breakdown field strength order of different samples was as follows: Micro-ZnO/LDPE < Micro-Nano-ZnO/LDPE < Nano-ZnO/LDPE. While the inorganic particles showed a better dispersion in polymers since the microparticles doping.

## Figures and Tables

**Figure 1 materials-13-01432-f001:**
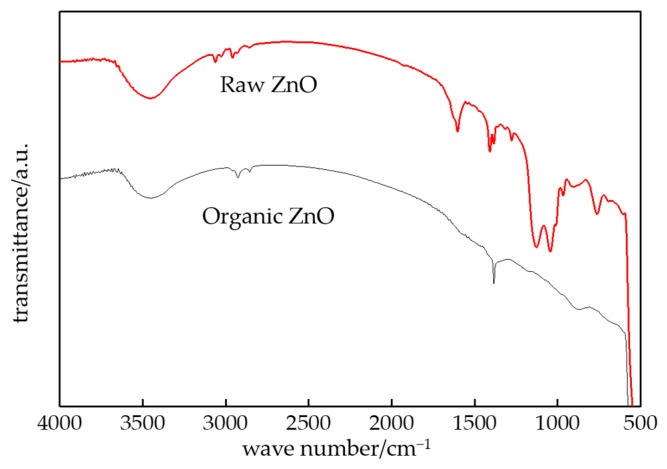
FTIR patterns of ZnO particles before and after surface modification.

**Figure 2 materials-13-01432-f002:**
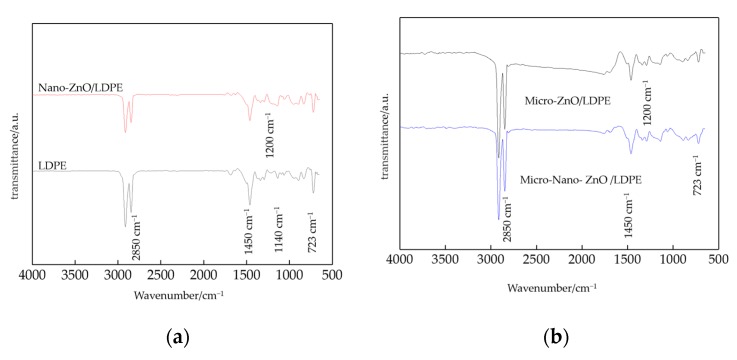
FTIR patterns of LDPE and different composites. (**a**) FTIR patterns of LDPE and Nano-ZnO/LDPE; (**b**) FTIR patterns of Micro-ZnO/LDPE, and Micro-/Nano-ZnO/LDPE. Note: 723 cm^−1^—C-H bond out-plane bending vibrational absorption peak; 1450 cm^−1^—C-H bond in-plane bending vibrational absorption peak; 2850 cm^−1^—C-H bond stretching vibrational absorption peak; 1200 cm^−1^—methylene symmetry and asymmetry low-intensity bending vibrational absorption peak; 1140 cm^−1^—C-H bond bending vibrational absorption peak.

**Figure 3 materials-13-01432-f003:**
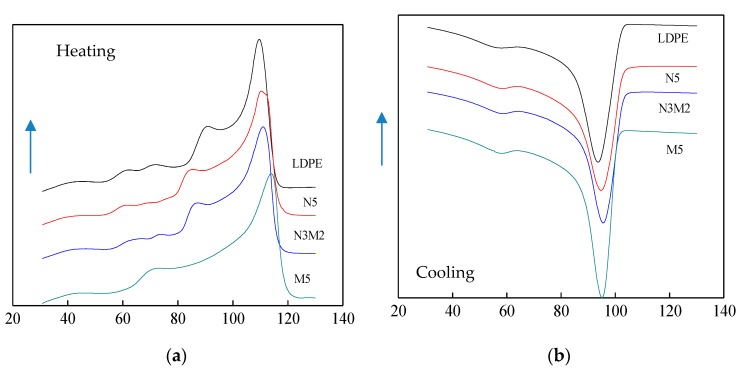
The heating and cooling curve of different samples. (**a**) Heating curve of different samples; (**b**) Cooling curve of different samples.

**Figure 4 materials-13-01432-f004:**
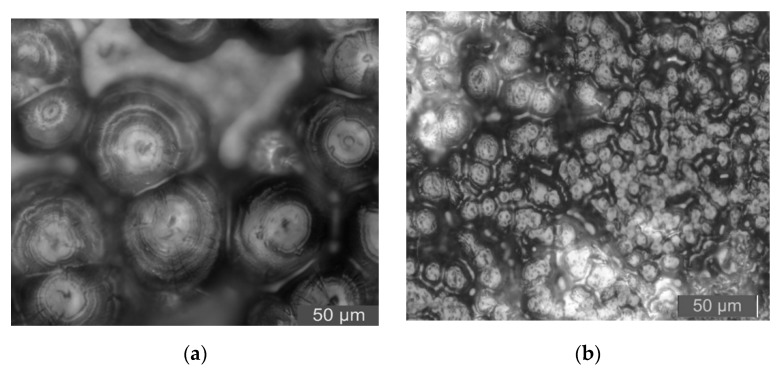
PLM pictures of Micro-Nano ZnO/LDPE composites. (**a**) LDPE; (**b**) N5; (**c**) M5; (**d**) N3M2.

**Figure 5 materials-13-01432-f005:**
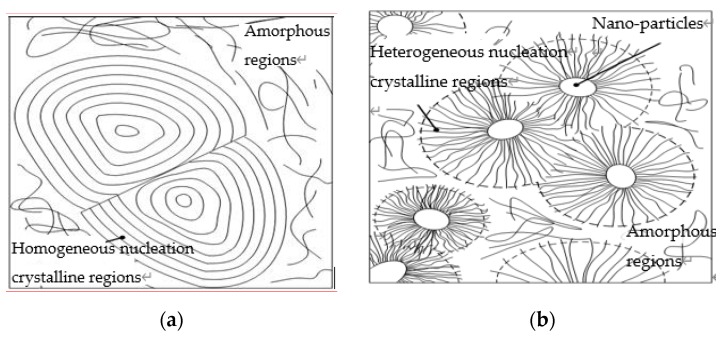
The structure model of Micro-Nano ZnO/LDPE composites. (**a**) LDPE; (**b**) N5; (**c**) M5; (**d**) N3M2.

**Figure 6 materials-13-01432-f006:**
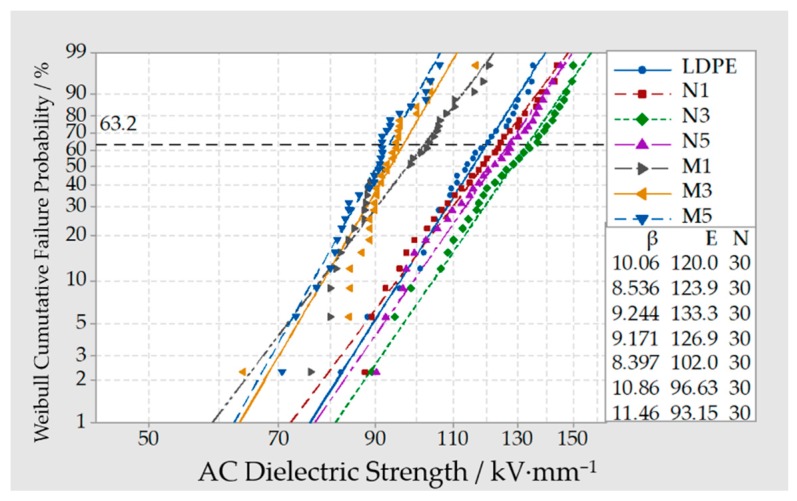
Weibull distribution curve of LDPE, different Micro-ZnO/LDPE, and Nano-ZnO/LDPE composites.

**Figure 7 materials-13-01432-f007:**
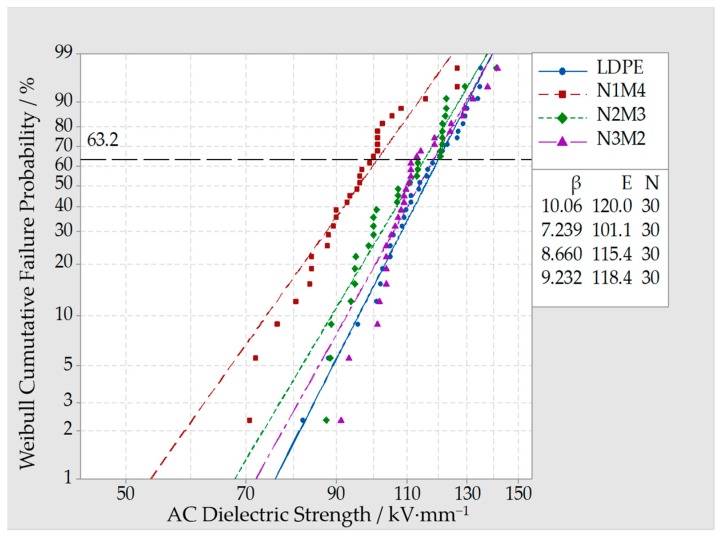
Weibull distribution curve of different Micro-Nano-ZnO/LDPE composites.

**Table 1 materials-13-01432-t001:** The name and filler content of the tested specimens.

Name	Model	Manufacturer
Torque rheometer	RM-200A	Harbin Hapro Electrical Technology Co., Ltd. (Harbin, Heilongjiang, China)
Plate vulcanization machine	XLB	Haimen Jinma Rubber & Plastics Machinery Technology Co., Ltd. (Haimen, Jiangsu, China)
Fourier transform infrared	EQUINOX55	Bruker (Karlsruhe, Baden-Wuerttemberg, Germany)
Polarization microscope	LeicaDM2500	Leica Microsystems Co., Ltd. (Frankfurt, Hesse-Darmstadt, Germany)
Differential scanning calorimeter	DSC-1	Mettler Toledo (Zurich, Zurich canton, Swizerland)
Ultrasonic cleaner	KQ5200DE	Kunshan Ultrasonic Instruments Co., Ltd. (Kunshan, Jiangsu, China)
Electric blender	JJ-1	Jincheng Guosheng Laboratory Instrument Work (Jincheng, Shanxi, China)
Thermostat water bath	DZKW-A	Shanghai Shuli Instrument and Meter Factory (Shanghai, China)
Vacuum oven	DZF-6020	Shanghai Boxun Industrial Co., Ltd. (Shanghai, China)
AC high voltage experimental console	JG-5	Shanghai Pujing Electrical Co., Ltd. (Shanghai, China)

**Table 2 materials-13-01432-t002:** The name and filler content of the tested specimens.

Sample	LDPE (wt%)	Mass Fraction of Nano-ZnO (wt%)	Mass Fraction of Micro-ZnO (wt%)
LDPE	100	0	0
N1	99	1	0
N3	97	3	0
N5	95	5	0
M1	99	0	1
M3	97	0	3
M5	95	0	5
N1M4	95	1	4
N2M3	95	2	3
N3M2	95	3	2

**Table 3 materials-13-01432-t003:** The melting peak and crystallinity of different samples.

Sample	T_c_ (°C)	T_con_ (°C)	ΔT_c_ (°C)	T_m_ (°C)	X_c_ (%)
LDPE	93.67	102.31	8.64	109.75	34.90
N5	95.15	102.65	7.50	109.84	36.18
N3M2	95.89	102.97	7.08	110.46	37.79
M5	95.65	100.66	5.03	113.66	36.01

Note: T_m_—The melting peak temperature; T_c_—The crystallization peak temperature; T_con_—initial crystallization temperature; T_c_—The width of Exothermic crystallization peak; X_c_—crystallinity.

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
