# Peer review of "The Research of Crystalline Morphology and Breakdown Characteristics of Polymer/Micro-Nano-Composites"

_materials, 2020, doi:10.3390/ma13061432_

Round 1

Reviewer 1 Report

The manuscript "The Research of Crystalline Morphology..." by Yujia Cheng and Guang Yu presents structural and electrical studies of ZnO particle/LDPE composites.

Firstly, the language must be addressed before publication can be considered. The grammatical and structural English issues are so significant as to impede reading of this manuscript. Large sections are unclear with respect to the authors meaning and intent. This must be addresses.

Secondly, there was a recent publication (Crystals 2019, 9, 481; doi:10.3390/cryst9090481) that addresses many of the topics in this manuscript. It considers ZnO nano and microparticle incorporation in LDPE, providing polarized microscopy images and breakdwon electrical curves that have the same conclusions as in the currently submitted manuscipt. The authors, at a minimum, must reference this work in their manuscript. But further, the authors need to indicate what new insights their work brings to this system. How does it compare to the previously published work? Is there any new or contrasting information here?

Thirdly, the authors need to define all acronyms the first time that they appear in the manuscript. This was neglected multiple times in the manuscript. Also, appropriate errors bars and statistical data need to be presented. For example, in Table 2 the crystallinity % values vary across the range by only +4%. Is that more than the observed variation in the results? the conclusions on the thermal data are all very reasonable, but the melting T variations are only +2%. It is reasonable (actually expected) that a mix of micro and nanoparticles will produce a better packing environment. But the thermal data show very small variations.

Author Response

Response to Reviewer 1 Comments:

Point 1: Firstly, the language must be addressed before publication can be considered. The grammatical and structural English issues are so significant as to impede reading of this manuscript. Large sections are unclear with respect to the authors meaning and intent. This must be addresses.

Response 1: According to the expert advice, the English language was revised carefully.

Point 2: Secondly, there was a recent publication (Crystals 2019, 9, 481; doi:10.3390/cryst9090481) that addresses many of the topics in this manuscript. It considers ZnO nano and microparticle incorporation in LDPE, providing polarized microscopy images and breakdwon electrical curves that have the same conclusions as in the currently submitted manuscipt. The authors, at a minimum, must reference this work in their manuscript. But further, the authors need to indicate what new insights their work brings to this system. How does it compare to the previously published work? Is there any new or contrasting information here?

Response 2: I want to explain about the why the publication in crystals was not referenced in my manuscript. We came from the same group. The crystalline morphology and dielectric properties of Micro-Nano-ZnO composite was my PhD research subject. The content of publication in crystals was the follow-up research. The emphasis in two articles were different, including two aspects:

(1) In their article, the space charge test was the main research content. But in our article, the main research content was the effect of Micro-Nano-ZnO particles doping on structure morphology and breakdown characteristics of polymer composite. According to PLM test, the crystalline morphology and size of different samples were observed. According to DSC test, the crystallization rate and crystallinity were explored. Based on the analysis of crystallization kinetics theory and combining the experiment result of PLM and DSC, the crystalline morphology model of Micro-Nano-ZnO/LDPE was built, and the mechanism of Micro-ZnO and Nano-ZnO particles doping to the crystallization properties of polymer was explored.

(2) The DC breakdown test was used in the publication of crystals, and the composites samples were only N3, M3 and N2M1. But in our manuscript, these samples were dealt with AC breakdown test. Although the matrix and filler were the same, more samples with different particles mass fraction were researched in our manuscript. The particles mass fraction were 1%, 3% and 5% respectively, including N1, N3, N5, M1, M3, M5, N1M4, N2M3, N3M2.

In summary, this manuscript was different with the previous publication. In our manuscript, according to the crystalline morphology test combining with AC breakdown test, the effect of different microstructure on macroscopic dielectric properties of polymer composite were explored.

Point 3: Thirdly, the authors need to define all acronyms the first time that they appear in the manuscript. This was neglected multiple times in the manuscript. Also, appropriate errors bars and statistical data need to be presented. For example, in Table 2 the crystallinity % values vary across the range by only +4%. Is that more than the observed variation in the results? the conclusions on the thermal data are all very reasonable, but the melting T variations are only +2%. It is reasonable (actually expected) that a mix of micro and nanoparticles will produce a better packing environment. But the thermal data show very small variations.

Response 3: According to the expert advice, all acronyms the first time that they appear in the manuscript were defined.

Besides, after different mass fraction particles doping, the melting temperature Tm of composites were higher than which of pure LDPE. Because The ZnO was a kind of inorganic materials, which provided good heat resistance. Besides, the inorganic particles provided excellent thermal conductivity, from which the heat would be transferred quickly. Therefore the melting temperature of composites were higher. According to the expert opinion, a mix of micro and nanoparticles will produce a better packing environment. Although the thermal data showed relatively small variations, the thermal data was reasonable. Since these equipment had high accuracy, the temperature error was ±0.1℃. Besides, the experimental results were verified repeatedly, and the data were consistent. The melting temperature of composites were all slightly higher than which of pure LDPE.

Reviewer 2 Report

The paper is generally well-written and interesting for the audience. However, I have few comments on the paper:

  1. Maybe authors can explain somehow, especially in the abstract what do they mean by micro-nano? This is a bit confusing in the abstract. What do mean by micro-nano in the same time for ZnO?
  2. In the Introduction: some grammar errors, like page 1, line 40 the polymer materials is... Please read the text carefully and correct all of the errors that have crept into the paper.
  3. Please provide more details on the chemical substrates, for example p-xylene - the purity etc.
  4. Authors should also provide more details on the equipment that has been applied: name, manufacturer...
  5. Page 7, line 210- something is weird with sentence "...composites was higher than which of pure LDPE". Maybe this can be rephrased.

Author Response

Response to Reviewer 2 Comments:

Point 1: Maybe authors can explain somehow, especially in the abstract what do they mean by micro-nano? This is a bit confusing in the abstract. What do mean by micro-nano in the same time for ZnO?

Response 1: In abstract, the “Nano-ZnO particles doping into LDPE” was defined as “Nano-ZnO/LDPE”, the “Micro-ZnO particles doping into LDPE” was defined as “Micro-ZnO/LDPE”, the “Nano-ZnO and Micro-ZnO particles doping into LDPE in the same time” was defined as “Micro-Nano-ZnO/LDPE”. All these were added into the abstract.

Point 2: In the Introduction: some grammar errors, like page 1, line 40 the polymer materials is... Please read the text carefully and correct all of the errors that have crept into the paper.

Response 2: According to the expert advice, the English language was revised carefully.

Point 3: Please provide more details on the chemical substrates, for example p-xylene - the purity etc.

Response 3: According to the expert advice, more details about the polymer materials were provided. For example, the P-xylene purity was 99%.

Point 4: Authors should also provide more details on the equipment that has been applied: name, manufacturer...

Response 4: According to the expert advice, more details on the equipment were provide, which were shown in table1.

Point 5: Page 7, line 210- something is weird with sentence "...composites was higher than which of pure LDPE". Maybe this can be rephrased.

Response 5: This sentence was rephrased as ”The melting peak temperature Tm of pure LDPE was lower than which of both Nano-ZnO/LDPE and Micro-ZnO/LDPE composites.”